# Developing a MySQL Database for the Provenance of Black Tiger Prawns (*Penaeus monodon*)

**DOI:** 10.3390/foods12142677

**Published:** 2023-07-11

**Authors:** Karthik Gopi, Debashish Mazumder, Jagoda Crawford, Patricia Gadd, Carol V. Tadros, Armand Atanacio, Neil Saintilan, Jesmond Sammut

**Affiliations:** 1Centre for Ecosystem Science, The School of Biological, Earth and Environmental Sciences, The University of New South Wales, Sydney, NSW 2052, Australia; 2University Centre for Rural Health, Sydney School of Public Health, The University of Sydney, Sydney, NSW 2052, Australia; 3Australian Nuclear Science and Technology Organisation (ANSTO), Sydney, NSW 2052, Australia; 4School of Natural Sciences, Macquarie University, Sydney, NSW 2052, Australia

**Keywords:** seafood, provenance, MySQL database, authentication, tiger prawn, elemental analysis, stable isotope analysis

## Abstract

As the demand for seafood increases, so does the incidence of seafood fraud. Confirming provenance of seafood is important to combat fraudulent labelling but requires a database that contains the isotopic and elemental “fingerprints” of authentic seafood samples. Local isotopic and elemental databases can be scaled up or combined with other databases to increase the spatial and species coverage to create a larger database. This study showcases the use of isotopic and elemental fingerprints of the black tiger prawn (*Penaeus monodon*) to develop a database that can be used to securely store the data necessary for determining provenance. The utility of this database was tested through querying and building seven different datasets that were used to develop models to determine the provenance of *P. monodon*. The models built using the data retrieved from the database demonstrated that the provenance of *P. monodon* could be determined with >80% accuracy. As the database was developed using MySQL, it can be scaled up to include additional regions, species, or methodologies depending on the needs of the users. Combining the database with methods of determining provenance will provide regulatory bodies and the seafood industry with another provenance tool to combat fraudulent seafood labelling.

## 1. Introduction

Fish, crustaceans, molluscs, and other aquatic animals are widely traded food commodities in the world, with a total farmgate sale value of USD 401 billion in 2018 and a record 179 million tons caught in the same year [1] The demand for seafood has increased globally at an average rate of 3.1% per annum from 1961 to 2017, driven by an increase in production as well as developments in processing, shipping, and distribution [1]. This, however, has increased the number of instances of seafood fraud across the globe [2,3,4,5] most acts of fraud are associated with poor or no monitoring at sea and ports and ineffective systems to determine provenance [6,7]. Cases of seafood fraud have been detected in countries such as Germany, Canada, the United States, the United Kingdom, Singapore, Taiwan, Australia, and New Zealand [2,3,4,5,8]. Many of these cases consist of mislabelling, where seafood is incorrectly labelled as a different species to raise the sale price and mislead the consumer [2,3,4,5,8]. For instance, Cawthorn [8] examined mislabelling in snappers (Family: Lutjanidae) in Canada, the United States, the United Kingdom, Singapore, Australia, and New Zealand and found that around 32% out of 300 samples were incorrectly named, and 40% were mislabelled. Similarly, an Oceana study examined 1215 samples collected across the United States, where around 33% of the samples were mislabelled [5]. Therefore, reliable methods of determining seafood provenance are necessary to act as a deterrent against seafood fraud.

Determining seafood provenance relies on DNA profiling, fatty acid profiling, various methods of elemental profiling (Inductively Coupled Plasma-Mass Spectrometry (ICP-MS), and Inductively Coupled Plasma-Atomic Emission Spectroscopy (ICP-AES)), stable isotope analysis (SIA), and micro-X-ray fluorescence (µXRF) through the Itrax XRF Core Scanner from Cox Analytical Systems [9]. Due to the complex supply chains of seafood products in the global market [10,11] recommended the use of multiple analytical methods to combat seafood fraud. Among them, the combined use of stable isotopes and elemental analysis discriminates between farmed and wild-caught samples and can separate geographic origins [12,13,14,15,16]. Previous studies have demonstrated that the combination of SIA and µXRF through Itrax provides >80% accuracy when determining seafood provenance [11,17]. The isotopic and elemental values from these analyses are used to test the provenance of unknown seafood samples using various classification and decision tree models [11,17].

Several studies have examined the use of larger regional or global scale databases to combat seafood fraud. For instance, Watson, Green [18] used global import and export databases to virtually track seafood through the supply chain. However, they noted that wild-caught or mariculture production was underestimated, and further work would be required to “ground-truth” or verify the provenance of seafood. Furthermore, the existing databases in countries like Europe focus on utilising DNA profiles of seafood [19,20]. Another database of note is the Barcode of Life Data System, which contains the DNA barcodes of 244k animal species [21] and has been successfully used to detect species mislabelling [22]. While DNA profiling is ideal for detecting species mislabelling, it has limited capacity to discriminate between the geographic origins of the same species or the production method [9]. Hence, a database of isotopic and elemental fingerprints would allow the provenance and production method of seafood to be determined accurately and reliably. However, unlike a DNA profile that does not change if a sample of the same seafood species is collected from a different location, the isotopic and elemental fingerprints can vary depending on location [11,17]. A realistic approach to development of a database of isotopic and elemental fingerprints would be to create localised databases that contain the isotopic and elemental values of commercially important species, and then combine these with other similar localised databases to build a wider coverage area. This would allow for flexibility in species and analytical methodology selection as researchers can collaborate with the local seafood industry to develop the database. To ensure that the data used for the databases are valid, the analyses should be conducted at certified laboratories.

Accordingly, the main aim of this paper is to lay the foundation for developing a localised database of isotopic and elemental fingerprints that can be expanded to include additional species or methodologies. Black tiger prawns (*Penaeus monodon*) were used as the test species for the proof of concept because the species is of commercial importance and has high production in the Asia Pacific [6,23]. The database can be easily expanded to include additional species or be created using a different species depending on a variety of factors such as conservation or commercial importance. The generalised hypotheses tested were:

(1) The isotopic and elemental values of *P. monodon* can be stored in a scalable database, which can be augmented with additional species or methods and used to determine provenance.

(2) The isotopic and elemental values of *P. monodon* stored in the database can be used to accurately determine provenance.

The completion of a localised Australian database of the isotopic and elemental values of *P. monodon* will allow for provenance to be determined accurately, and act as a point of comparison for unknown samples of *P. monodon*. Ensuring that the database is scalable will also guarantee that it can be kept up to date and allow for additional species and analyses to be included, thereby acting as a useful tool to combat seafood fraud and protect the local seafood industry. There is also the potential for the database to be scaled regionally and worldwide at a later stage.

## 2. Materials and Methods

### 2.1. Sample Collection

To ensure that authentic black tiger prawn samples were collected from the market, the sampling was undertaken with the assistance of Sydney Fish Market (SFM) staff with a good knowledge of the supply chain; they were able to authenticate where seafood was farmed or wild caught. The samples were collected from batches of black tiger prawns sent to the market for sale from the Eastern Seaboard of Australia (ESA) over 2 years (January–March 2020–2021). As the samples were collected from the same season in both years, we did not control for seasonal variability. Nine individual *P. monodon* samples from each of the 5 farms and 3 wild-catch locations from the ESA were collected over 2 years (2020–2021) (Figure 1). Producers from these locations regularly shipped produce to the SFM, and products were traceable back to their origins. Environmental interference was not controlled for as it led to the differences between origins [11,17].

The length of each individual *P. monodon* sample was measured to ensure that samples from each location were of similar size to minimise ontogenetic effects. The samples were all roughly equal in length (~16–20 cm) and came from batches of seafood sent to the market for sale. The samples were frozen (−20 °C) and transported to the Australian Nuclear Science and Technology Organisation (ANSTO) for processing and analyses [11,17]. No live *P. monodon* samples were used.

### 2.2. Sample Preparation

The samples were then thoroughly rinsed with de-ionised water to remove any surface contaminants before the head, shell, and hindgut were removed. A 5 cm^2^ sample of abdominal tissue was then removed and rinsed with de-ionised water again before being oven-dried at 60 °C for 48 h. Once dry, the samples were ground into a fine powder for homogeneity, which was used for all subsequent analyses [11,17].

### 2.3. Elemental Profiling

An Itrax micro-X-ray fluorescence (µXRF) core scanner with a molybdenum tube was used to determine the relative elemental abundance (given as a percentage of the total counts of elements) of the *P. monodon* samples. For a more detailed overview of the methods, refer to Gadd [24]. XRF through Itrax was chosen, as it was used in previous studies and produced accurate results when determining provenance [11,17].

To complement the use of Itrax, particle induced X-ray emission (PIXE) and particle induced Gamma-ray emission (PIGE) ion beam analysis (IBA) techniques were used to determine if utilising the elemental concentration of *P. monodon* would improve accuracy when predicting provenance. These IBA techniques are commonly used in air pollution studies [25,26,27,28] to determine the elemental source fingerprints and to apportion fine airborne particulate matter samples. IBA techniques are non-destructive and can measure elemental concentrations down to 1 µg/g in a few minutes of accelerator beam-time. The PIGE IBA technique was used to detect elements between lithium (Li) and aluminium (Al), whereas the PIXE technique, performed simultaneously with PIGE, was used to detect elements from aluminium (Al) to uranium (U).

### 2.4. Stable Isotope Analysis

The stable carbon and nitrogen isotopic analyses were conducted at ANSTO in New South Wales, Australia, using a continuous-flow isotope ratio mass spectrometer (CF-IRMS) model Delta V Plus (Thermo Scientific Corporation, Waltham, MA, USA), interfaced with an elemental analyser (Thermo Fisher Flash 2000 HT EA, Thermo Electron Corporation, Waltham, MA, USA). The obtained isotopic values were relative to the International Atomic Energy Agency (IAEA) secondary standards and were certified relative to Vienna-PeeDee Belemnite (VPDB) for carbon and air for nitrogen. The data were normalised and quality controlled using the standards Chitin and Caesin Sodium Salt from Bovine Mil, which bracketed the analysed samples. The results were accurate to 1% for both C% and N% and ±0.3 parts per thousand (‰) for δ^13^C and δ^15^N. They were reported in delta (δ) units in parts per thousand (‰) determined by the formula:$$X\left(‰\right)=\left(\frac{Rsample}{Rstandard}-1\right)\times 1000$$

While the lipid contents in muscles could affect the δ^13^C values of crustaceans [29,30], all the C:N ratios of the samples used in this study were below 3.5, and, therefore, the δ^13^C values did not need to be mathematically corrected.

### 2.5. Data Management and Statistical Analyses

#### 2.5.1. Database Development

To store the large dataset obtained from µXRF through Itrax, IBA, and SIA, a database was created using MySQL [31]. MySQL is an open-source database management system based on Structured Query Language (SQL). A database built using MySQL can be encrypted, contain many different types of data that can be queried and joined using statements, and can be scaled to include up to 50 million records [32]. MySQL is effectively used by companies like Facebook and Netflix to manage large databases [32]. As the aim of the study is to lay the foundation for developing a localised database of isotopic and elemental fingerprints that can be expanded, the database should be built up using software that can efficiently store and retrieve data from large databases. Users can efficiently query specific entries, filter using keywords, and link the database with programming software, such as R, for analysis.

Having an open-source database management system ensures that the methodology could be made accessible to all researchers and end users. The database served as the basis for all subsequent analyses and was used to compare the values when predicting provenance. As the values were specific to each farm, the dataset will currently only be made available to the commercial entities and researchers involved in this project to protect their confidentiality.

#### 2.5.2. Mapping the Sampling Locations

To allow the end user to easily examine the values of the different farms, a map of the locations included in the database was created using raster, ggplot2, rgdal, and rgeos [33,34,35,36] in R v4.1.1 [37].

#### 2.5.3. Determining Provenance

To test the ability of the database to predict provenance, the database was separated into training and test datasets. Due to the available sample size, the test dataset took a subset of one sample from every location, and the training dataset was made up of the remaining samples. The test and training datasets were then used to train and test the accuracy of random forest and linear discriminant analysis. Two different models were tested, random forest and LDA, to ensure that the database could be retrieved and analysed in R. The differences between the models were not the major focus of this study. Users of the database could utilise any of the models available to determine provenance.

##### Random Forest

Random forest is an ensemble learning method [38] in R [37], where many different decision trees are generated from the data. These trees use the isotopic and elemental fingerprints of each “testing” or unknown provenance sample and uses multiple decision trees to determine provenance. The trees generated by random forest are by default randomised, and a seed must be set to ensure that results are reproducible. We used a seed of 1234 and random forest models with 500 trees. This model has been used in previous studies and has typically provided >80% accuracy when predicting provenance [11,17].

##### Linear Discriminant Analysis

Linear discriminant analysis (LDA) is a dimensionless model that is commonly used in provenance studies [11,17,39] and finds a combination of the variables in the dataset to obtain the best separation between different groups. The isotopic and elemental values of the samples in each group is examined and grouped in a way to reduce the separation within each origin while maximising the distance between different groups. The LDA function from the MASS package in R was used to discriminate between the farmed and wild-catch locations [40].

## 3. Results

### 3.1. Developing the Database

The isotopic and elemental values of the *P. monodon* samples from five farms and three wild-catch locations were stored in comma-separated values (CSV) files by the three laboratories that conducted the analyses. In total, from nine replicates from each location, there were 4128 values made up of the C:N ratio and δ^13^C and δ^15^N values from stable isotope analysis, the PPM values of F, Na, Al, Si, P, S, Cl, K, Ca, Ti, V, Cr, Mn, Fe, Co, Ni, Cu, Zn, As, Br, Rb, Sr, Y, and Zr from IBA, and the counts of Mg, Al, Si, P, S, Cl, K, Ca, Ti, Cr, Mn, Fe, Ni, Cu, Zn, As, Se, Br, Rb, Sr, Y, Zr, Cd, Sn, Sb, Nd, Hf, Pb, Bi, At, and U from µXRF through Itrax. These values, along with additional metadata, were stored in a relational database built using MySQL, which consisted of linked tables (Figure 2). The database allowed for data to be filtered and retrieved using queries, which becomes more important as the database grows.

### 3.2. Determining the Provenance Using the Database

The isotopic and elemental fingerprints stored in the database were extracted using various queries to build seven different datasets (Table 1). These datasets were then used to train the random forest and LDA models. The model accuracy was calculated from the number of incorrect classifications by the model from the training dataset. Higher model accuracy meant that more samples were correctly classified to their origin by the model. The accuracy of the random forest model was further tested by using a testing subset. These samples were withheld from the training dataset, and then the model was asked to predict the origin. Figure 3 demonstrates the process used by the trained random forest model to determine the provenance of prawn samples from unknown origins.

There are some geographic overlaps between sampling locations in the map (Figure 1), but these overlaps did not affect the provenance models because random forest and LDA could distinguish between the overlapping origins using their isotopic and elemental profiles. There are some differences between the accuracies of random forest and LDA used in this study. The full set of results from random forest and LDA are provided in Table 1. Datasets 1 through to 5 performed well but had quite a few incorrect predictions when determining the origins of the testing dataset. Although Dataset 6 had the highest accuracy for both random forest and LDA (~89% and ~69%, respectively), Dataset 7 had a fewer number of incorrect predictions (Table 1). The accuracy of LDA models, apart from Dataset 6, dropped once the number of variables increased.

## 4. Discussion

The analytical techniques used to determine the isotopic and elemental fingerprints of tiger prawn (*P. monodon*) samples from operational farms and wild-catch fisheries along the ESA resulted in the acquisition of large and comprehensive datasets (Section 3). Therefore, a relational database management system using MySQL Server was developed to collectively manage (store and update) the data efficiently, along with the capabilities to retrieve and analyse the data using a variety of analysis software like R, Python, Julia, or Matlab. The ability to link the database with software like R allows users to quickly apply models, such as random forest, to determine seafood provenance. As the database is designed to enable end-users to authenticate samples from a particular origin, representative black tiger prawn samples authenticated by the Sydney Fish Market provided a solid starting point for the database. As distinct fingerprints are created for authentic *P. monodon* samples from other farms along the Eastern seaboard of Australia, the database will be updated (Figure 2) and serve as a national databank of *P. monodon* for provenance evaluation. Due to commercial sensitivity, the fingerprint data will currently only be released to commercial entities and researchers involved in this phase of the research effort. However, in the future, the data may also be released to academic researchers and industry for scientific collaboration, where data access will be initiated through application, and access will be granted based on data usage requirements. This strategy can be extended to develop an extensive national reference database for commonly traded seafood species (e.g., snapper, barramundi, bluefin tuna), thereby providing the Australian seafood industry with authenticated data to reliably verify the provenance of seafood products and ensure the correct labelling of tiger prawn products. The foundation laid out here can be applied to other regions and species of seafood with relative ease. The MySQL 8.0 Reference Manual provides the documentation necessary to get started with MySQL, and users are welcome to set up databases in a manner that benefits them [32]. If databases are structured well, structured query language can be used to manipulate and combine different databases together.

Here, it is demonstrated that this database is a vital tool for provenance model applications. A sample of *P. monodon* that is of unknown origin can be compared to this database using random forest or LDA to determine its production method and geographic origin from the locations included in the database. The random forest models are more robust and typically have better accuracy than the LDA models, for which too many variables were attached to a relatively small sample size (see dataset 2, 4 and 7; Table 1). Given that the model training accuracy was less than 50% in most cases for LDA, random forest is the preferred model for this dataset. A larger dataset that is more representative of the source regions would be expected to perform better on test samples. Oliveri [41] mentioned that data for each modelled class needs to be “sampled in a fully representative way”. This would require extensive sampling, additional testing of the models using validation, and thus ensuring a representative model. The incorrect predictions in this study could well be due to the training data not being fully representative. Hence, when the withheld samples are tested against the model, it cannot predict their origins accurately. We are currently testing the efficacy of various models in determining seafood provenance using a larger dataset. Researchers have used various models to determine provenance, and the models used here are only to demonstrate the utility of having an easily accessible and scalable database. In previous studies, combining SIA and µXRF through Itrax gave the best results when predicting provenance [11,17]. While the combination of SIA and Itrax had the highest accuracy in this study, it also had more incorrect predictions than a combination of all three analytical techniques. Datasets 4, 5, 6, and 7, built using random forest, show the most promise, as they have high accuracy with minimal incorrect predictions. However, as the number of locations increase, dataset 7, with further calibration, such as using only selected variables or different parameters, is likely to have less incorrect predictions when testing provenance.

There have been several studies that demonstrated the utility of elemental profiling to trace seafood back to its origin. The majority of these studies use laser-ablation inductively coupled mass spectrometry (LA-ICP-MS) and have focused on testing whether this can be used to determine seafood provenance on a number of different species [14,15,16]. These studies had an accuracy of between 68–85% when determining the origin of blue mussels (*Mytilus edulis*), green-lipped mussels (*Perna canaliculus*), and Southern Keeled Octopi (*Octopus berrima*). Similarly, isotopic fingerprinting has been used extensively in seafood provenance research, with varying degrees of success [9,12,13]. Using stable isotopes alone to determine the provenance of finfish and prawns leads to around 50–100% accuracy. However, studies typically combine stable isotopes with other analytical methods such as DNA or elemental profiling to ensure minimal incorrect provenance predictions [9].

While random forest and LDA can accurately determine the provenance of *P. monodon*, they heavily depend on an accurate and reliable reference database of isotopic and elemental values for comparison. Ideally, the isotopic and elemental values of seafood from all major origins would be catalogued in an open-access database that can be accessed by researchers, regulatory bodies, the seafood industry, and consumers across the globe. However, this would be a difficult task, as research is currently scattered across multiple analytical techniques and species. Therefore, the ideal solution to tackle the issue of seafood provenance is to create localised databases that focus on commercially valuable species of seafood. This is still a significant undertaking but will have the potential to link with other localised databases to cover a wider geographical area and multiple species. This study was aimed at laying the foundation for creating such a database of the major production areas of *P. monodon* along the ESA. The database contains the data gathered using SIA, µXRF through Itrax and IBA, making it unique, as it contains data from multiple analyses and from locations that typically supply consumers along the ESA. The database created using MySQL also has the potential to be scaled up alongside the database to handle additional species or analyses as required [42]. Trusted parties can be given secure access to the database to contribute the fingerprints from other species or analyses. Ensuring that the database is updated regularly will allow for it to be a vital tool to combat the growing threat of seafood fraud. Furthermore, this concept can be applied to samples from any region, and, in the future, it can also be combined with emerging methodologies like blockchain to securely store and transmit data to end users [9]. This would also allow for contributions from other localised databases to overcome the issue of creating a centralised database of isotopic and elemental values of seafood to combat seafood fraud.

Isotopic and elemental data can be used to determine seafood provenance with a relatively high degree of accuracy. However, this approach relies on having a database of accurate values from known origins to compare unknown samples against. Dataset 7 (SIA, IBA and µXRF through Itrax) has high accuracy with minimal incorrect predictions and is likely to be more robust than dataset 5 (IBA and SIA), as the number of locations in the database increase. The newly developed localised database from this study, created using MySQL, digitises information on provenance and elemental signatures determined from authentic black tiger prawns (*P. monodon)* from the Eastern Seaboard of Australia. The database can be expanded to incorporate databases from additional geographic origins and species. Future studies can also explore the use of other database management systems if MySQL is not fit for purpose. Combining both the database and methods of determining provenance will provide regulatory bodies and the industry with the scientific tools needed to verify seafood provenance.

## Figures and Tables

**Figure 1 foods-12-02677-f001:**
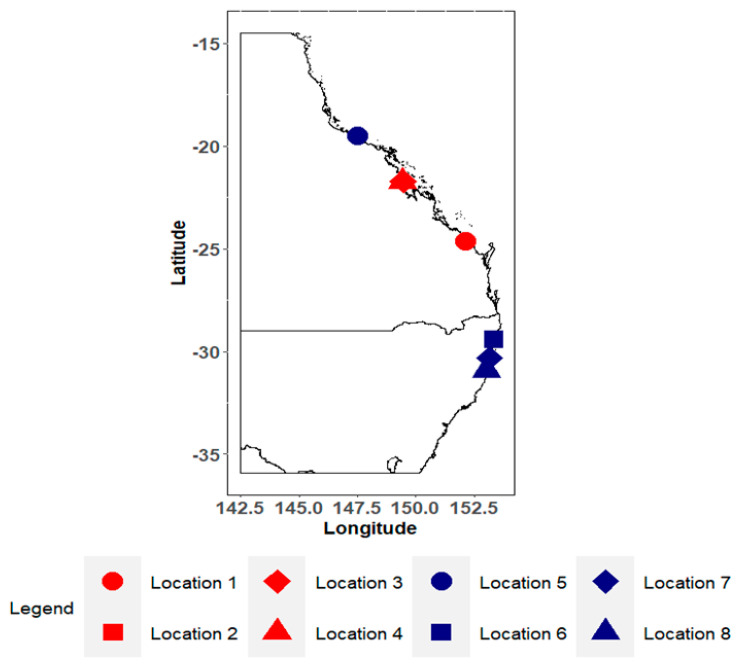
Locations where *P. monodon* samples were collected from; red locations represent farms, and blue represents wild−catchment areas. Location 2 and 6 are geographically close but represent different production methods (wild-caught vs. farmed).

**Figure 2 foods-12-02677-f002:**
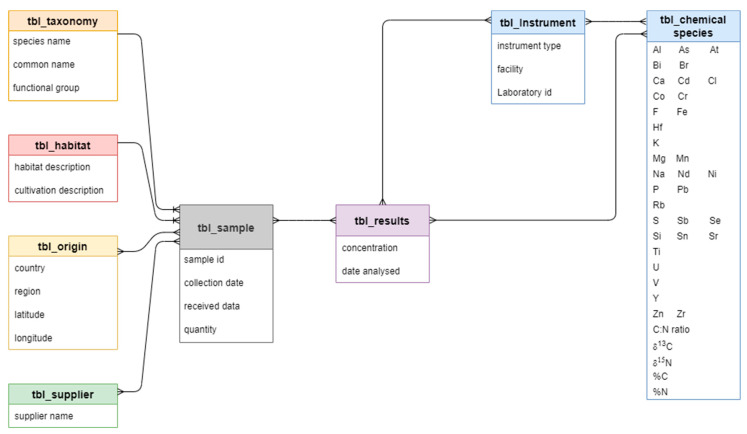
The structure of the tiger prawn (*P. monodon*) database, showing core tables and the relationships between them.

**Figure 3 foods-12-02677-f003:**
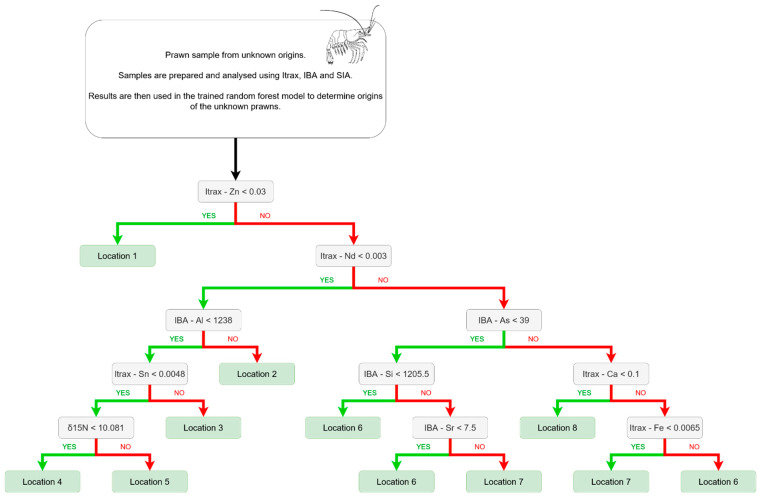
An example of a decision tree that is used by a random forest model to determine provenance. A yes or no decision is made at each branch of the tree until the sample is sorted into an origin. For instance, the first element it looks at is Zinc detected by Itrax; if the value is below 0.03, then the sample is assigned to “Source A”. If the value is greater than 0.03, then it branches over to look at Neodymium detected by Itrax, and so on until it reaches an origin. This process is repeated with 500 different trees to determine the origin of the test sample.

**Table 1 foods-12-02677-t001:** Summary of the percent accuracies of random forest and LDA models created using various datasets, and the percent of incorrect predictions when the models were tested.

		Random Forest		LDA
Number of Variables		Model Accuracy	Prediction Accuracy	Model Accuracy
Dataset 1: IBA	24	73.33%	74.07%	49.37%
Dataset 2: Itrax	31	77.78%	78.57%	43.98%
Dataset 3: SIA	3	63.74%	82.14%	50.20%
Dataset 4: IBA + Itrax	55	77.53%	88.89%	36.44%
Dataset 5: IBA + SIA	27	87.78%	81.48%	49.94%
Dataset 6: Itrax + SIA	34	88.89%	89.29%	68.54%
Dataset 7: IBA + Itrax + SIA	58	87.64%	92.59%	47.55%

## Data Availability

The datasets presented in this manuscript are not readily available due to commercial-in-confidence. Requests to access the datasets should be directed to the corresponding author.

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
