# Peer review of "Developing a MySQL Database for the Provenance of Black Tiger Prawns (Penaeus monodon)"

_foods, 2023, doi:10.3390/foods12142677_

Round 1
Reviewer 1 Report
The author proposed a MySQL database for identifying the provenance of P. monodon, which was established by measuring isotopes and elements from real samples. The core idea of the manuscript is to identify P. monodon with MySQL database to avoid incidents of seafood fraud. The starting point of this manuscript is of practical significance, but some details need to be further provided. Therefore, this manuscript needs major revisions.
Please find below some specific comments:
1. How efficient is the search of MySQL database? How does the search performance compare to other similar databases?
2. How do test results compare between multiple test sets? How to accurately distinguish between different test sets with similar test samples?
3. Have you compared testing and training results between small and large datasets? How consistent and stable is it?
4. How to eliminate the environmental interference during sample collection and database test training?
Reviewer 2 Report
This manuscript develops a database based on the isotopic and elemental fingerprints for determining the provenance of black tiger prawn. The topic is meaningful. However, the method is obsolescence, The collected data are limited. The writing is lack of logic. Therefore, I do not recommend publishing this manuscript.
Reviewer 3 Report
The paper titled “Developing a MySQL database for the provenance of black tiger prawns (Penaeus monodon)” presents vividly the necessity, the way of building, and the verification of a database for the authentication of black tiger prawns based on the isotopic and elemental prevalence of a large number of elements. The subject is interesting, the presentation is well documented, and the use of English flawless.
However, despite the fact that the authors invite other researchers to use the database and contribute with data from other localised databases, it would be very important for readers to present in greater detail the trees that lead to the identification of the samples.
Furthermore, possible explanations for the wrong identifications during the tests would be of great importance both for the reader as well as a troubleshooting guide for the authors themselves in order to improve the efficiency of the database.
Reviewer 4 Report
This paper entilted « Developing a MySQL database for the provenance of black tiger prawns (Penaeus monodon) » on the use of a MySQL database is a first step for the implementation of a geographical origin tool to trace seafood product alongside Australia coasts. This study was conducted with care, which is essential for establishing a baseline.
The comparison of the two statistical treatments of the data from the different elemental and isotopic analyses to evaluate the accuracy of prediction of the origin of the Tiger shrimp random forests and linear discriminant analysis (LDA) samples is very interesting. It seems to me that the authors may not have insisted enough on the interest of using random forests which give more accurate results. In case of dispute, the use of LDA does not seem relevant given the number of variables and reference samples in the database, since the result given by this treatment has only about a one in two chance of being accurate.
I’ve no specific comments except line 165, I suppose that it’s « Bovine milk » and not « Bovine Mil ».
Round 2
Reviewer 1 Report
Accept as it is
Author Response
No comments from reviewer 1, but tick boxes showed that the Methods and Results can be improved.
Response:
Figure 2 from the methods section has been moved into the results section to better explain how the random forest model determines the provenance of prawn samples from unknown origins. Further improvements have also been undertaken as detailed in the responses to reviewer 2 and 3.
Reviewer 2 Report
The authors failed to answer the question. The quality of manuscript has not improved significantly. Simply speaking, regardless of innovation and the fullness of data, the figures and tables should at least be neat and bequtiful.
Author Response
Comment:
The authors failed to answer the question. The quality of the manuscript has not improved significantly. Simply speaking, regardless of innovation and the fullness of data, the figures and the tables should at least be neat and beautiful. manuscript.
Response:
We moved Figure 2 from the methods section to the results section (Figure 2 becomes Figure 3 now) to explain the results instead of the conceptual version of decision-making trees. We also included a sentence explaining new Figure 3 in the results section (Page 7, lines 257-258, red colour). Thus, Figure 3 is redrawn (Page 8) to make it more meaningful and relevant to this study. Similarly, Table 1 was amended to make it clean and nicer (Page 9, red colour).
Reviewer 3 Report
Although the authors comments clarified issues of the manuscript I also now find the paper less interesting, mainly because (a) there is no connection presented between the decision trees and the actual chemical / biological findings nor to the environmental conditions and (b) an overfitted model resembles a self-fulfilling prophecy, which indicates that further and better training of the model would be required.
Author Response
Comment:
Although the authors comments clarified issues of the manuscript, I also now find the paper less interesting, mainly because (a) there is no connection presented between the decision trees and the actual chemical/biological findings not to the environmental conditions and (b) an overfitted model resembles a self-fulfilling prophecy, which indicates that further and better training of the model would be required.
Response:
(a) We created a new Figure 3 to align with decision trees used by random forest model to determine provenance. Figure 3 (it was Figure 2 before) now has been moved from methods to the results section to better represent a decision tree used in this application, and is logically aligned with both methods and results. The caption of Figure 3 has also been re-written to provide a clear explanation of connections between branches of the trees pointing towards a decision (Page 8, Figure 3 Lines 260-264, red colour).
(b) Here we were trying to indicate that overfitting is an issue when insufficient data is available. Basically, the training can only be undertaken on what data is available and predictions of samples that differ from the training set can potentially be incorrect. We rephrased the following sentences and included them in the discussion section.
“This would require extensive sampling, additional testing of the models using validation and thus ensuring a representative model. The incorrect predictions in this study could well be due to the training data not being fully representative.” (Page 10, Lines 325-327, red colour).